# Pigmentation and TYRP1 expression are mediated by zinc through the early secretory pathway-resident ZNT proteins

Takumi Wagatsuma[1], Eisuke Suzuki[1], Miku Shiotsu[1], Akiko Sogo[2], Yukina Nishito[1], Hideya Ando[3], Hisashi Hashimoto[4], Michael J. Petris[5,6,7], Masato Kinoshita ● [2] & Taiho Kambe ● [1✉]

Tyrosinase (TYR) and tyrosinase-related proteins 1 and 2 (TYRP1 and TYRP2) are essential for pigmentation. They are generally classified as type-3 copper proteins, with binuclear copper active sites. Although there is experimental evidence for a copper cofactor in TYR, delivered via the copper transporter, ATP7A, the presence of copper in TYRP1 and TYRP2 has not been demonstrated. Here, we report that the expression and function of TYRP1 requires zinc, mediated by ZNT5–ZNT6 heterodimers (ZNT5–6) or ZNT7–ZNT7 homodimers (ZNT7). Loss of ZNT5–6 and ZNT7 function results in hypopigmentation in medaka fish and human melanoma cells, and is accompanied by immature melanosomes and reduced melanin content, as observed in TYRP1 dysfunction. The requirement of ZNT5–6 and ZNT7 for TYRP1 expression is conserved in human, mouse, and chicken orthologs. Our results provide novel insights into the pigmentation process and address questions regarding metalation in tyrosinase protein family.

[1] Division of Integrated Life Science, Graduate School of Biostudies, Kyoto University, Kyoto 606-8502, Japan. [2] Division of Applied Biosciences, Graduate School of Agriculture, Kyoto University, Kyoto 606-8502, Japan. [3] Department of Applied Chemistry and Biotechnology, Okayama University of Science, Okayama 700-0005, Japan. [4] Division of Biological Science, Graduate School of Science, Nagoya University, Nagoya, Aichi 464-8602, Japan. [5] Departments of Ophthalmology, University of Missouri, Columbia, MO 65211, USA. [6] Biochemistry, University of Missouri, Columbia, MO 65211, USA. [7] Bond Life Sciences Center, Columbia, MO 65211, USA. ✉email: kambe.taiho.7z@kyoto-u.ac.jp

Melanin is the main pigment responsible for the color of the skin, hair, and iris[1,2]. Its biosynthesis requires three melanogenic enzymes, tyrosinase (TYR) and tyrosinase-related proteins 1 and 2 (TYRP1 and TYRP2), all of which are localized within melanosomes. In humans, mutations in *TYR* result in oculocutaneous albinism (OCA) type 1[3], whereas mutations in *TYRP1* or *TYRP2* give rise to OCA type 3[4] or OCA type 8[5], respectively. In mice, functional impairment of Tyr, Tyrp1, and Tyrp2 results in changes in the coat color from black to white, brown, and dark gray, respectively[6–8]. These data indicate the crucial role of these proteins in melanogenesis. TYR is the rate-limiting enzyme in melanin pigmentation because it catalyzes the hydroxylation of tyrosine to L-3,4-dihydroxyphenylalanine (DOPA) and the oxidation of DOPA to dopaquinone[2,9]. Dopaquinone is then converted to dihydroxyindole carboxylic acid (DHICA) via transformation of dopachrome by TYRP2 (dopachrome tautomerase)[10]. DHICA is oxidized to indole-5, 6 quinone-carboxylic acid, which polymerizes to yield eumelanin. TYRP1 is thought to function as a DHICA oxidase, however, its role in the enzymatic reaction is the subject of debate in the field[11,12].

TYR and TYRP1 share ~40% amino acid sequence identity and are generally classified as type-3 copper proteins that are characterized by a binuclear copper active site[2,13,14], based on the crystal structure of TYR from mushrooms and other organisms[15,16]. In the binuclear copper active site, each copper ion is coordinated to three conserved histidine (His) residues[2,13,14]. The insertion of copper ions into TYR occurs during its biosynthesis in the secretory pathway, which is mediated by the ATP7A copper transporter (Menkes copper transporter) located in the *trans*-Golgi network (TGN) and melanosomes[17–19]. Thus, copper metalation is essential for the maturation of TYR. In contrast, the identity of the metal in the active site of TYRP1 and the mechanism of metalation is not clear with some reports suggesting that zinc may be a cofactor in TYRP1[20–23].

If indeed zinc, rather than Cu, occupies the active site of TYRP1, melanogenesis would depend on transporters that supply zinc within secretory compartments during TYRP1 biosynthesis. In vertebrate cells, two zinc transporter family proteins, Zn transporters (ZNTs) and Zrt, Irt-like protein (ZIPs), play pivotal roles in transporting zinc across cellular membranes[24,25], and previous studies have shown that Zn transporters, specifically the ZNT5-6 heterodimer and the ZNT7 homodimer, supply zinc to the lumen of the early secretory compartments, including the endoplasmic reticulum (ER) and Golgi apparatus[25–27]. Both ZNT5-6 and ZNT7 are required to metallate a number of zinc-requiring enzymes that function in the intracellular compartments, on the plasma membrane, or extracellularly by supplying zinc to them[28–33]. In this study, we examined the effect of disrupting the Zn transporters ZNT5 and ZNT7 (because ZNT6 loses zinc transport activity and functions as the auxiliary subunit of ZNT5-6[34,35]) on pigmentation in human melanoma cells and medaka fish, and provide evidence that ZNT5–6 and ZNT7 are essential for TYRP1 expression and activity.

## Results

**Impairment of Znt5–6 and Znt7 function results in the formation of immature melanosomes in medaka fish.** To understand the role of Znt5-6 and Znt7 in melanogenesis, we generated $Znt5^{-/-};Znt7^{-/-}$ mutant medaka larvae using CRISPR-Cas9[33]. Whereas both wild type (WT) and $Znt5^{+/-};Znt7^{-/-}$ medaka larvae at 8–9 days post-fertilization (dpf) exhibited darkly pigmented black spots (Fig. 1a, b), the pigmented spots in the homozygous

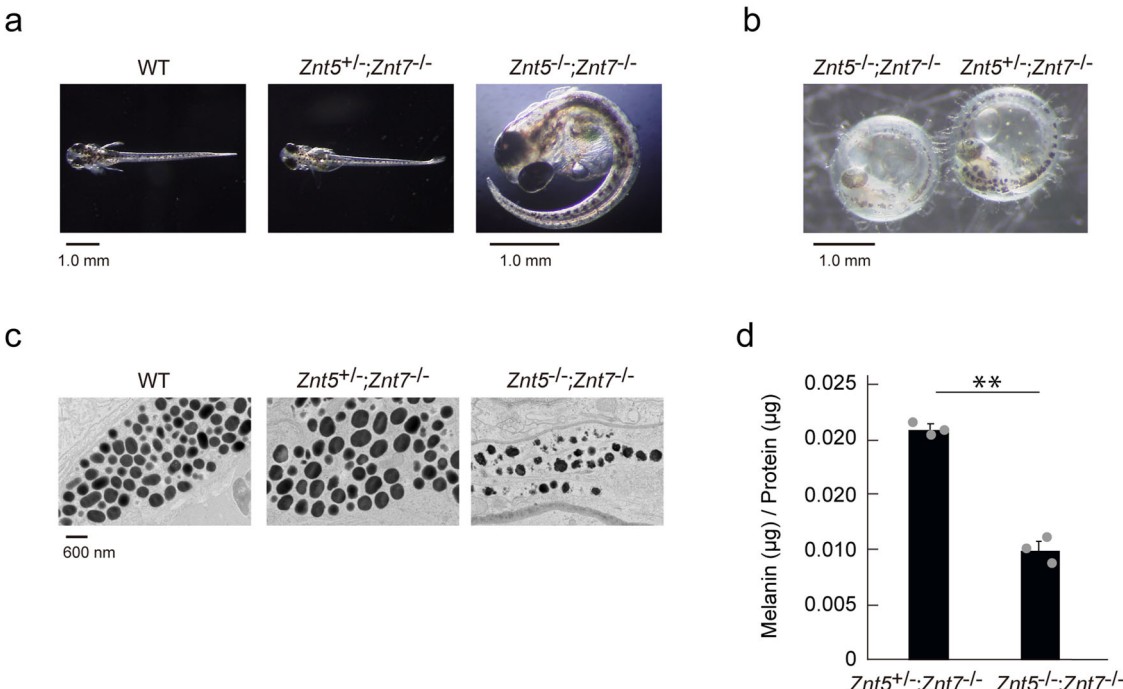

**Fig. 1 Znt5-6 and Znt7 are required for melanogenesis in medaka fish. a** Representative dorsal views of whole larvae at 8–9 dpf for wild-type (WT), $Znt5^{+/-}$;$Znt7^{-/-}$, and $Znt5^{-/-}$;$Znt7^{-/-}$ medaka. The $Znt5^{-/-}$;$Znt7^{-/-}$ medaka did not hatch, and thus is shown after manual removal of the chorion. **b** Representative lateral views of embryos at 8–9 dpf for $Znt5^{-/-}$;$Znt7^{-/-}$ (*left*) and $Znt5^{+/-}$;$Znt7^{-/-}$ (*right*) medaka before hatching. **c** Transmission electron microscopy (TEM) shows irregular melanosomes with less electron dense pigment in $Znt5^{-/-}$;$Znt7^{-/-}$ medaka compared with those in WT and $Znt5^{+/-}$;$Znt7^{-/-}$ medaka. **d** Melanin content was decreased in $Znt5^{-/-}$;$Znt7^{-/-}$ mutant medaka compared with that in $Znt5^{+/-}$;$Znt7^{-/-}$ littermate ($n = 3$). Statistical significance was analyzed by Student's *t*-test. **$p < 0.01$. In **a–c**, each analysis was performed on more than three individual medaka, and in **d**, the experiments were performed at least three times, and representative results are shown.

$Znt5^{-/-}$;$Znt7^{-/-}$ mutant larvae appeared smaller and were lighter in color, suggesting an impairment in melanogenesis. To further evaluate this phenotype, we performed ultrastructural analysis of melanosomes using transmission electron microscopy (TEM). As expected, the melanosomes of WT and $Znt5^{+/-}$;$Znt7^{-/-}$ medaka (8–9 dpf) appeared round with well-circumscribed borders and were densely packed. In contrast, the melanosomes in $Znt5^{-/-}$;$Znt7^{-/-}$ mutant medaka were smaller, less densely packed and had irregular and granular borders (Fig. 1c). Consistent with these defects in melanosome morphology, the melanin content in $Znt5^{-/-}$;$Znt7^{-/-}$ mutant medaka was significantly lower than that in $Znt5^{+/-}$;$Znt7^{-/-}$ littermate medaka (Fig. 1d). Therefore, these data confirm that simultaneous loss of Znt5–6 and Znt7 results in defective melanin synthesis in medaka melanocytes.

**ZNT5–6 and ZNT7 play crucial roles in melanin synthesis in human melanoma cells**. Next, we examined whether ZNT5–6 and ZNT7 dysfunction would cause a similar hypopigmented phenotype in a human melanoma Mewo cell line. After 10 days in culture, Mewo cells produce sufficient melanin such that cell pellets appear dark brown in color (Fig. 2a). We generated ZNT5 and ZNT7 double-knockout Mewo cells (Mewo-Z5Z7-DKO cells) by disrupting ZNT5 and ZNT7 using CRISPR/Cas9 gene editing (Table 1). The color of pelleted Mewo-Z5Z7-DKO cells was a lighter shade of brown compared to WT Mewo cells (Fig. 2a), and contained considerably less melanin content (Fig. 2b). Importantly, these defects in Mewo-Z5Z7-DKO cells were reversed by re-expression of ZNT7; specifically, re-expression of ZNT7 in Mewo-Z5Z7-DKO cells reversed the light brown color of the pelleted cells to dark brown (Fig. 2a) and restored the melanin content to a level comparable with that in WT Mewo cells (Fig. 2b). Similar to the mutant medaka fish, TEM analysis of Mewo-Z5Z7-DKO cells revealed an increase in immature and irregularly shaped melanosomes compared to WT Mewo cells (Fig. 2c). These results indicate that ZNT5–6 and ZNT7 contribute to melanin synthesis and that their functions are conserved in vertebrates.

**ZNT5–6 and ZNT7 are essential for the expression of TYRP1**. $Tyrp1$-deficient and dominant-negative Tyrp1-expressing zebrafish show hypopigmented brown spots and irregular morphology of melanosomes[36,37], similar to those found in $Znt5^{-/-}$;$Znt7^{-/-}$ mutant medaka (Fig. 1). Moreover, $Tyrp1$-deficient animals show a hypopigmented coat color, and their melanocytes contain less electron dense and irregular melanosomes[8,38–40] that appear to phenocopy those of Mewo-Z5Z7-DKO cells (Fig. 2). Such similarities prompted us to test if the pigmentation defects caused by the loss of ZNT5–6 and ZNT7 might be attributable to defects in TYRP1.

Immunoblotting and immunofluorescence analyses showed that the expression of TYRP1 was substantially decreased in Mewo-Z5Z7-DKO cells (Fig. 3a, b). In contrast, we found no differences in the expression of TYR and TYRP2 in Mewo-Z5Z7-DKO cells compared to WT Mewo cells (Fig. 3a, b). Stable re-expression of

ZNT7 in Mewo-Z5Z7-DKO cells restored TYRP1 expression (Fig. 3c, d), which is consistent with the ZNT7-dependent restoration of melanin synthesis seen earlier (Fig. 2a, b). Decreased TYRP1 expression was not restored by zinc supplementation

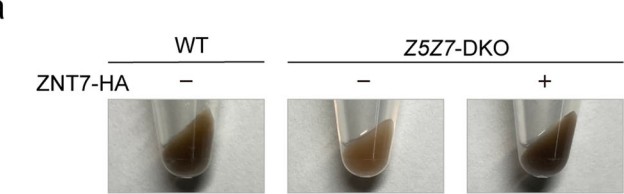

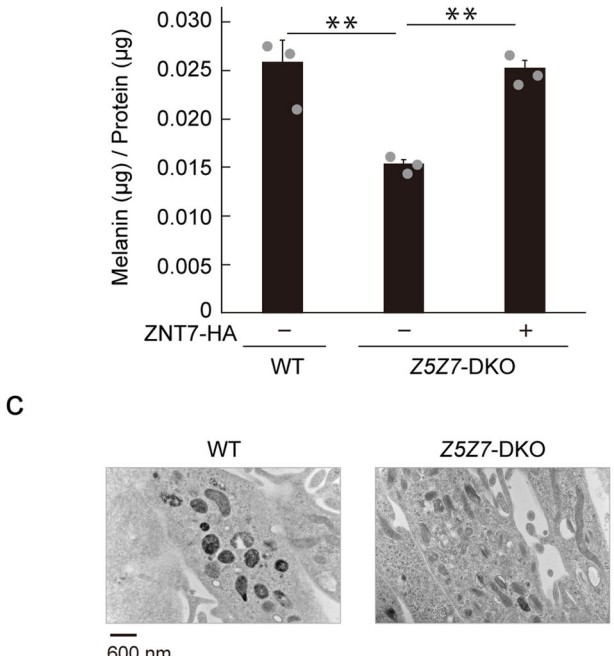

**Fig. 2 Melanin synthesis is decreased in Mewo-Z5Z7-DKO cells. a** Pellets of wild-type (WT) Mewo and Mewo-Z5Z7-DKO cells, and of Mewo-Z5Z7-DKO cells stably re-expressing ZNT7-HA. **b** Melanin content was decreased in Mewo-Z5Z7-DKO cells compared with that in WT Mewo cells; the decrease was recovered upon expression of ZNT7 ($n = 3$). Statistical significance was determined using the one-way analysis of variance (ANOVA) followed by Tukey's post hoc test. **\*\***$p < 0.01$. **c** Transmission electron microscopy (TEM) showing irregular melanosomes with less electron dense pigment in Mewo-Z5Z7-DKO cells compared with those in WT Mewo cells. Each experiment, except for that in **c**, was performed at least three times, and representative results from independent experiments are shown. Panels in **c** show representative results for at least three cells.

| Cells | Targeted region | Mutation |
|---|---|---|
| Mewo-Z5Z7-DKO | Exon 11 in ZNT5, exon 2 in ZNT7 | 5 bp deletion, 17 bp deletion, 91 bp insertion in ZNT5, 5 bp deletion in both alleles of ZNT7 |
| Mewo-TYRP1-KO | Exon 7 | 37 bp insertion in both alleles of TYRP1 |
| SK-ATP7A-KO | Exon 8 | 2 bp deletion in both alleles of ATP7A |
| SK- Z5Z7ATP7A-TKO | Exon 11 in ZNT5, exon 2 in ZNT7, exon 8 in ATP7A | 13 bp or 18 bp deletion in ZNT5, 1 bp or 7 bp insertion in ZNT7, 2 bp deletion in both alleles of ATP7A |

**Table 1 Mutations introduced in ATP7A, TYRP1, ZNT5, and ZNT7 in cells used in this study.**

Information of SK-Z5Z7-DKO is referred in a previous study[33].

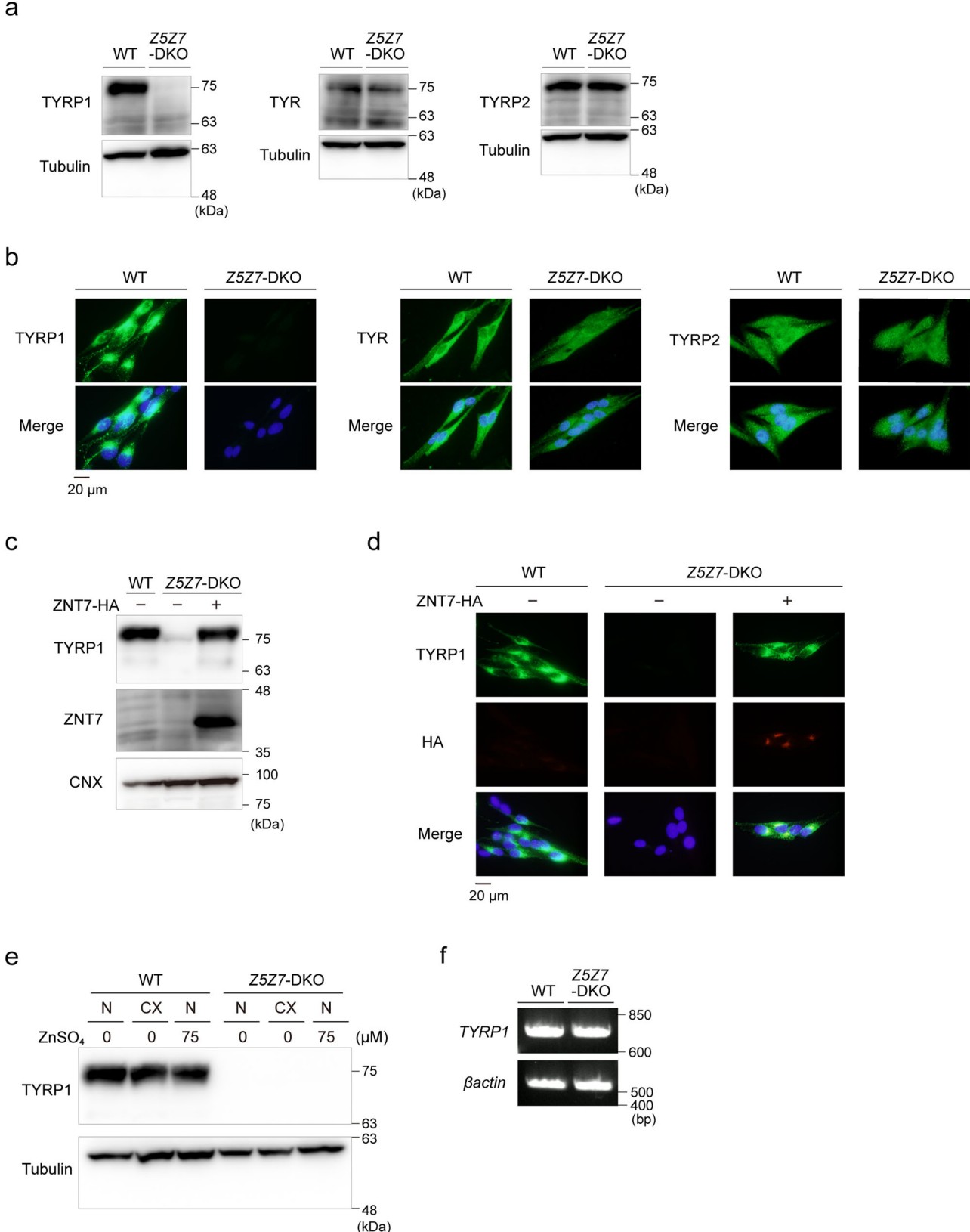

**Fig. 3 TYPR1 expression is substantially decreased in Mewo-*Z5Z7*-DKO cells. a**, **b** Expression of TYRP1, but not TYR or TYRP2, was substantially decreased in Mewo-*Z5Z7*-DKO cells. Immunoblotting (**a**) and immunofluorescence staining (**b**). **c**, **d** Expression of TYRP1 in Mewo-*Z5Z7*-DKO cells was reversed by stable re-expression of ZNT7. Immunoblotting (**c**) and immunofluorescence staining (**d**). **e** Effects of zinc status on the expression of TYRP1. Wild-type (WT) Mewo and Mewo-*Z5Z7*-DKO cells were cultured in normal medium (N), normal medium supplemented with ZnSO$_4$ (75 μM), or in zinc-deficient medium (CX) generated using Chelex-treated FCS for 24 h. **f** *TYRP1* mRNA levels were not decreased in Mewo-*Z5Z7*-DKO cells. In **a**, **c**, and **e**, Tubulin or CNX was detected as a loading control. Each experiment was performed at least three times, and representative results from independent experiments are shown.

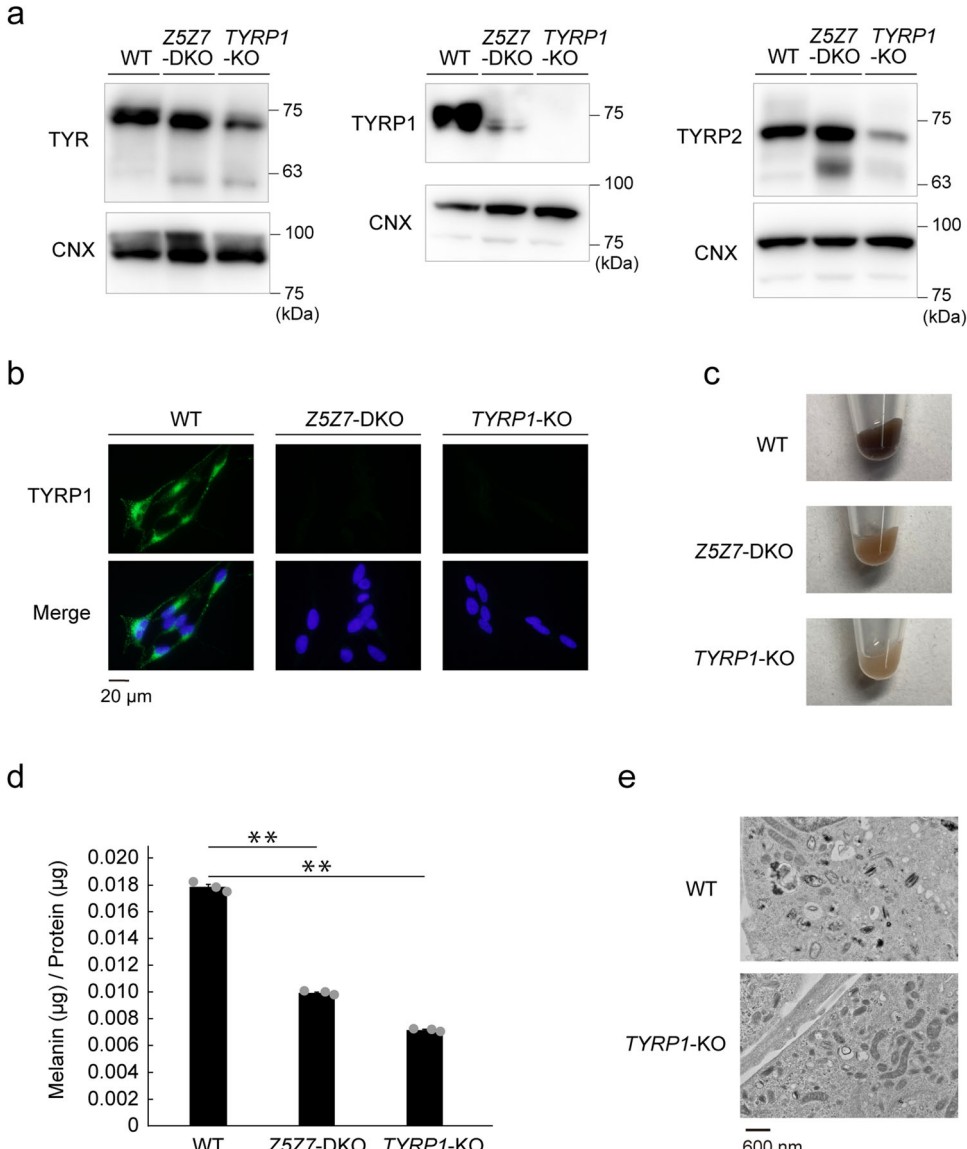

**Fig. 4 Mewo-*Z5Z7*-DKO cells show similar defects as in Mewo-*TYRP1*-KO cells. a** Analysis of TYRP1, TYR, and TYR2 expression by immunoblotting in wild-type (WT) Mewo, Mewo-*Z5Z7*-DKO and Mewo-*TYRP1*-KO cells. Note the loss of TYRP1, but not TYR and TYRP2, in both Mewo-*Z5Z7*-DKO and Mewo-*TYRP1*-KO cells. **b** Immunofluorescence analysis confirmed a loss of TYRP1 in both Mewo-*Z5Z7*-DKO and Mewo-*TYRP1*-KO cells. **c** Pellets of Mewo-*Z5Z7*-DKO and Mewo-*TYRP1*-KO cells exhibit hypopigmentation compared to WT Mewo cells. **d** Melanin content was decreased in Mewo-*TYRP1*-KO cells, similar to that in Mewo-*Z5Z7*-DKO cells ($n = 3$). Statistical significance was determined using the one-way analysis of variance (ANOVA) followed by Tukey's post hoc test. **$p < 0.01$. **e** TEM showed irregular melanosomes with less electron dense contents in Mewo-*TYRP1*-KO, compared with WT Mewo cells. Each experiment except for (**e**) was performed at least three times, and representative results from independent experiments are shown. Panels in **e** were performed using at least three cells, and representative results are shown.

(75 μM ZnSO₄; Fig. 3e). Moreover, zinc deficiency, which was generated by an ~100-fold reduction in zinc concentrations by treating fetal calf serum (FCS) with Chelex resin[41], did not decrease TYRP1 expression (Fig. 3e), suggesting that the effect of zinc, mediated by ZNT5–6 and ZNT7, is crucial and specific. The mRNA expression levels of *TYRP1* were not significantly different between Mewo-*Z5Z7*-DKO and WT Mewo cells (Fig. 3f), suggesting that apo-TYRP1 may be destabilized in Mewo-*Z5Z7*-DKO cells.

We generated Mewo cells deficient in *TYRP1* (Mewo-*TYRP1*-KO cells) using CRISPR/Cas9 gene editing (Table 1) and compared their phenotypes with those of Mewo-*Z5Z7*-DKO cells (Fig. 4a, b). The pelleted Mewo-*TYRP1*-KO cells showed a light brown color similar to that of Mewo-*Z5Z7*-DKO cells (Fig. 4c), and the melanin content was also similar (Fig. 4d). Similar to Mewo-*Z5Z7*-DKO cells (Fig. 2c), TEM analysis revealed many

immature melanosomes of reduced electron density in Mewo-*TYRP1*-KO cells (Fig. 4e). These results strongly suggested that ZNT5–6 and ZNT7 are crucial for the expression of TYRP1.

**Detailed examination of the roles of ZNT5–6 and ZNT7 in TYRP1 expression.** We examined how ZNT5–6 and ZNT7 are associated with the expression of TYRP1 in another human melanoma cell line, SK-MEL-2, which is deficient in both *ZNT5* and *ZNT7* (SK-*Z5Z7*-DKO), because of their usefulness in transient transfection studies[33]. We transiently transfected expression plasmids of TYR, TYRP1, and TYRP2 into WT SK-MEL-2 cells and SK-*Z5Z7*-DKO cells and examined their expression. Expression of TYR, TYRP1, and TYRP2 was confirmed in WT SK-MEL-2 cells using immunoblotting and immunofluorescence staining. Consistent with the results obtained in Mewo cells, the

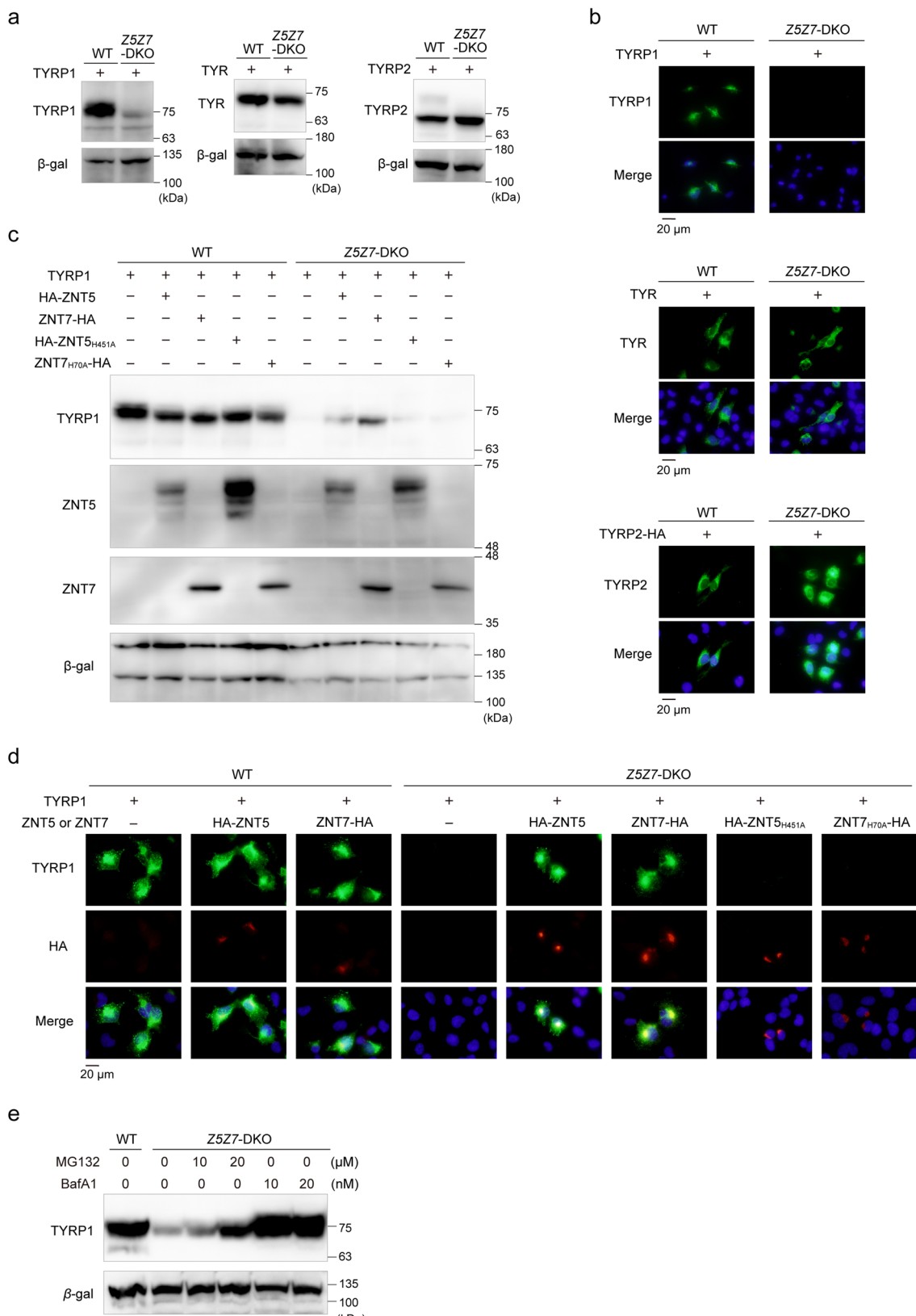

expression of TYRP1 protein was substantially decreased in SK-*Z5Z7*-DKO cells, even when expressed from a strong promoter (Fig. 5a, b), which is in contrast to the unaltered expression observed for TYR and TYRP2 in these cells, although the upper band in the immunoblots of TYRP2 was lost in SK-*Z5Z7*-DKO cells (Fig. 5a). The expression of TYRP1 in SK-*Z5Z7*-DKO cells was restored by the simultaneous expression of ZNT5 or ZNT7 (Fig. 5c, d). However, it was not restored by zinc transport-incompetent ZNT5 and ZNT7 mutants (ZNT5H451A or ZNT7H70A)[28,29] (Fig. 5c, d). The substantial decrease in the expression of transfected TYRP1 in SK-*Z5Z7*-DKO cells was restored by bafilomycin A1 treatment, suggesting that TYRP1 is

**Fig. 5 TYRP1 expression is dependent on functional ZNT5 or ZNT7 transporters. a** Immunoblot analysis of TYRP1, TYR, and TYRP2 in wild-type (WT) SK-MEL-2 cells and SK-Z5Z7-DKO cells transfected with plasmids encoding TYRP1, TYR and TYRP2. Note that TYRP1, but not TYR and TYRP2, was substantially decreased in SK-Z5Z7-DKO cells. **b** Confirmation of substantial reduction in TYRP1 expression using immunofluorescence staining. **c** Immunoblot and **d** immunofluorescence microscopy showing restoration of TYRP1 expression in SK-Z5Z7-DKO cells following re-expression of wild type ZNT5 or ZNT7, but not of zinc transport-incompetent mutants of ZNT5 and ZNT7 (ZNT5_{H451A} and ZNT7_{H70A}). **e** Bafilomycin A1 (BafA1) treatment stabilizes the expression of TYRP1, transiently transfected in SK-Z5Z7-DKO cells. SK-Z5Z7-DKO cells were treated with the indicated concentrations of MG132 and BafA1 for 6 h. In **c**, **d**, TYR, TYRP1, and TYRP2 plasmids and ZNT plasmids were transfected at a ratio of 1:10. In **a**, **c**, and **e**, β-galactosidase (β-gal) was used as a transfection control. Each experiment was performed at least three times, and representative results from independent experiments are shown.

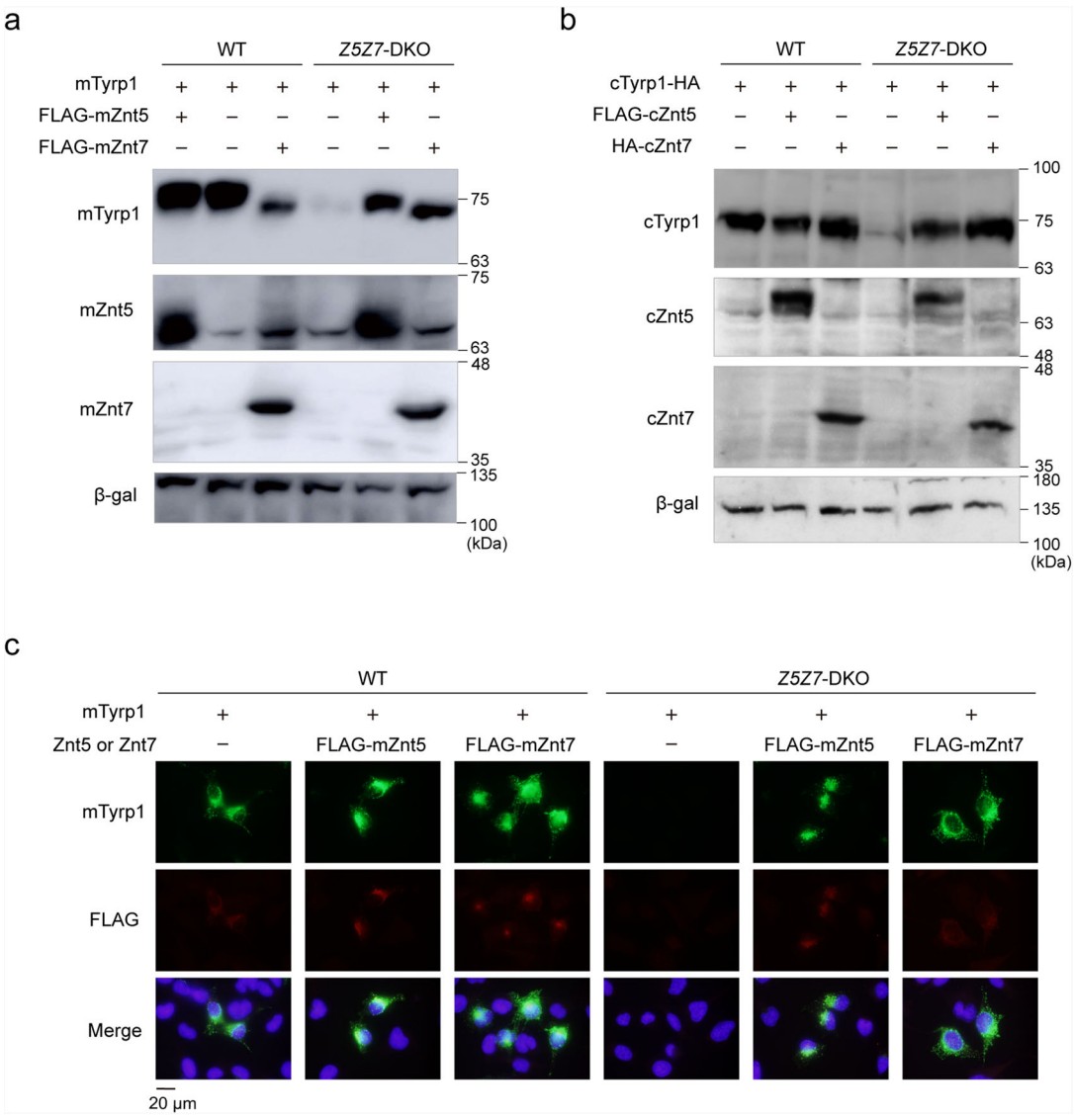

**Fig. 6 Restoration of mouse or chicken Tyrp1 expression in SK-Z5Z7-DKO cells following re-expression of mouse or chicken Znt5 or Znt7.**
**a**, **b** Expression of recombinant mouse (**a**) and chicken (**b**) Tyrp1 was substantially decreased in SK-Z5Z7-DKO cells compared to wild-type (WT) SK-MEL-2 cells, but stabilized by expression of recombinant mouse or chicken Znt5 or Znt7. In **b**, cTyrp1, tagged with HA at the C-terminus, was used. **c** Immunofluorescence staining of mouse Tyrp1 expressed in WT SK-MEL-2 or SK-Z5Z7-DKO cells. Tyrp1 and Znt plasmids were transfected at a ratio of 1:10. Each experiment was performed at least three times, and representative results from independent experiments are shown.

degraded in the lysosomal degradation pathway in DKO cells (Fig. 5e), although it was partly degraded in the proteasomal pathway because MG132 treatment minimally restored its expression (Fig. 5e).

Human TYRP1 and mouse Tyrp1 have different functions in the DHICA oxidase reaction[11,12]. Thus, we examined whether the dependence of TYRP1 on ZNT5–6 and ZNT7 in humans is conserved for Tyrp1, Znt5–6, and Znt7 in mice using transient transfection in SK-Z5Z7-DKO cells. Substantially reduced expression of mouse Tyrp1 was observed in transiently transfected SK-Z5Z7-DKO cells (Fig. 6a), as observed for human TYRP1. The reduced expression of Tyrp1 was restored by

simultaneous expression of mouse Znt5 or Znt7 (Fig. 6a). The same Znt5- or Znt7-dependency of Tyrp1 was observed for their chicken counterparts (Fig. 6b). Moreover, a substantial reduction in the expression of mouse Tyrp1 and its restoration upon re-expression of mouse Znt5 or Znt7 was also confirmed using immunofluorescence staining (Fig. 6c). These results indicate that the requirement of ZNT5–6 and ZNT7 for TYRP1 expression is an intrinsic property of TYRP1.

**ATP7A is required for TYR activity but not for TYRP1 expression.** We generated SK-MEL-2 cells deficient in *ATP7A* (SK-*ATP7A*-KO cells) using CRISPR/Cas9 gene editing (Table 1) to confirm that the loss of the TGN-resident copper transporter, ATP7A, does not affect the expression of TYRP1. We transiently transfected TYR, TYRP1, and TYRP2 into SK-*ATP7A*-KO cells and examined their expression levels. TYRP1 was detected in SK-*ATP7A*-KO cells at levels comparable to those in WT SK-MEL-2 cells, similar to TYR and TYRP2 (Fig. 7a). Consistent with a previous report in human fibroblast cells[17], the activity of TYR (L-DOPA oxidase activity and tyrosinase zymography) was substantially decreased in SK-*ATP7A*-KO cells (Fig. 7b), indicating that ATP7A is indispensable for TYR activity and that TYR is present as an *apo*protein in SK-*ATP7A*-KO cells. To further clarify the different dependencies of TYR and TYRP1 on ATP7A, ZNT5–6, and ZNT7, we established SK-MEL-2 cells deficient in *ATP7A*, *ZNT5*, and *ZNT7* (SK-*Z5Z7ATP7A*-triple knockout (TKO) cells) (Table 1). The expression of TYRP1 in transiently transfected SK-*Z5Z7ATP7A*-TKO cells was substantially decreased, similar to that in SK-*Z5Z7*-DKO cells (Fig. 7c). Moreover, the activity of TYR in SK-*Z5Z7ATP7A*-TKO cells was lost, similar to that in SK-*ATP7A*-KO cells (Fig. 7d). These results clearly indicate that the biological functions of TYR and TYRP1 are differently regulated by metals, mediated by specific metal transporters—TYRP1 expression requires zinc and is mediated by ZNT5–6 and ZNT7, whereas TYR requires copper and is mediated by ATP7A.

TYR and TYRP1 have >70% similarity, but sequence alignment revealed a major sequence difference in the C-terminal portion containing the membrane-spanning region between TYR and TYRP1. We constructed TYR-_TYRP1_ or TYRP1-_TYR_ chimeras in which the C-terminal portion of TYR was substituted with that of TYRP1, and vice versa in the latter (Fig. 7e), and transiently expressed them in WT SK-MEL-2, SK-*ATP7A*-KO, and SK-*Z5Z7*-DKO cells. The TYR-_TYRP1_ chimeras were expressed in all the transfected cells at comparable levels, as was TYR (Fig. 7f). TYRP1-_TYR_ chimera showed a substantial reduction in WT SK-MEL-2 cells and SK-*ATP7A*-KO cells, compared with TYRP1 (Fig. 7f). It also showed a decrease in expression similar to that of TYRP1 in SK-*Z5Z7*-DKO cells (Fig. 7f), suggesting that the sequence responsible for the decreased expression of TYRP1 in SK-*Z5Z7*-DKO cells may be present in the core domain of TYRP1 (Met1 to Ala456) located in the melanosomes.

## Discussion

The role of ATP7A in supplying copper to TYR is evidenced by the hypopigmentation phenotype resulting from functional loss of ATP7A in zebrafish[42,43] and mice[44–46]. While such studies have defined a critical role for copper in melanogenesis, the role of zinc in melanogenesis is less clear. In this study, we provide evidence that melanogenesis is dependent on zinc transporters ZNT5–6 and ZNT7, which are known to supply zinc to nascent metalloenzymes within early secretory compartments. Our finding that loss of ZNT5–6 and ZNT7 in both medaka and human melanoma cells results in defective melanogenesis suggests this requirement is conserved in vertebrates. Our results suggest that

hypopigmentation could be attributed to the loss of TYRP1 expression, which was caused by impaired zinc supply, mediated by ZNT5–6 and ZNT7. To the best of our knowledge, this is the first study to directly implicate the importance of both zinc and copper transporters in melanin synthesis.

TYRP1 is categorized as type 3 copper protein, along with TYR and TYRP2, and is believed to acquire two copper ions during biosynthesis in the secretory pathway. However, whether TYRP1 is metalated by copper remains controversial[20–23]. Our results clearly show that the expression of TYRP1 is dependent on functional ZNT5–6 and ZNT7, but not on inactive mutants of these transporters, indicating that TYRP1 is stabilized by Zn. Furthermore, we found no evidence that TYRP1 expression was stabilized by copper transport via ATP7A. Consistent with our results, a recent X-ray structure of TYRP1 revealed two zinc ions at the binuclear metal-binding active site[22,23]. Therefore, these findings should help resolve the debate on the metal specificity of TYRP1 and support a model in which TYRP1 activity and stability requires binuclear zinc in its active site that is delivered into secretory compartments by ZNT5–6 and ZNT7. Furthermore, our results suggest that the failure to metallate TYRP1 due to the impairment of ZNT5–6 and ZNT7 results in the degradation of TYRP1 via a lysosomal dependent pathway. Such a model is reminiscent of the instability of apo-alkaline phosphatases we have reported previously, which is also caused by the loss of ZNT5–6 and ZNT7[28,31,33]. The results of our study, together with previous findings that TYRP2 contains two zinc ions in the active site[20,47,48], suggest that the assignment of TYRP1 and TYRP2 as type 3 copper proteins should be amended. In the immunoblot analysis in this study, the upper band of TYRP2 that was dependent on ZNT5 or ZNT7 expression may reflect the zinc-bound conformation of TYRP2 or possibly a complex with these zinc transporters. A similar ZNT5-6 and ZNT7 dependency was noted for a slower migrating band of carbonic anhydrase IX in a previous study[29]. However, more studies are needed to understand the mechanisms by which TYRP2 acquires zinc during the secretory process.

While the requirement for TYRP1 in melanin synthesis is undisputed, certain questions remain regarding its enzymatic functions. TYRP1 is assumed to function as a DHICA oxidase[49], based on the detection of DHICA oxidase activity in mouse Tyrp1[11], although for reasons that are not clear, human TYRP1 does not have such activity[12]. Our findings reveal that the ZNT5–6 and ZNT7 are indispensable for the expression of TYRP1 orthologs in human, mouse, and chicken, which argues that TYRP1 itself is unlikely to catalyze the oxidation of DHICA because zinc has no redox properties. It has been suggested that the DHICA oxidase function of TYRP1 could be achieved by forming hetero-complexes with TYR which contains redox active copper[50,51]. An alternative possibility is that nascent TYRP1 is first metalated with zinc via ZNT5–6 and ZNT7 in early secretory compartments, and subsequently displaced by copper in the TGN or melanosomes by ATP7A. It will be interesting to clarify this possibility in future studies.

Several in vitro kinetic and structural studies have indicated that TYR activity is inhibited by zinc[52–54]. TYR and TYRP1, as well as TYRP2, are highly homologous, which raises the question of how metal selectivity between copper and zinc is achieved at their active sites. An unknown specific mechanism may operate for metalating copper at the active site of TYR or zinc at the active site of TYRP1. Different routes of intracellular trafficking of TYR and TYRP1, as suggested previously[55–57], may provide a mechanism that allows these proteins to be appropriately metallated by copper and zinc. We initially predicted that the cytoplasmic C-terminal portion containing the transmembrane domain of TYRP1 might be important in this regard because this region has less homology

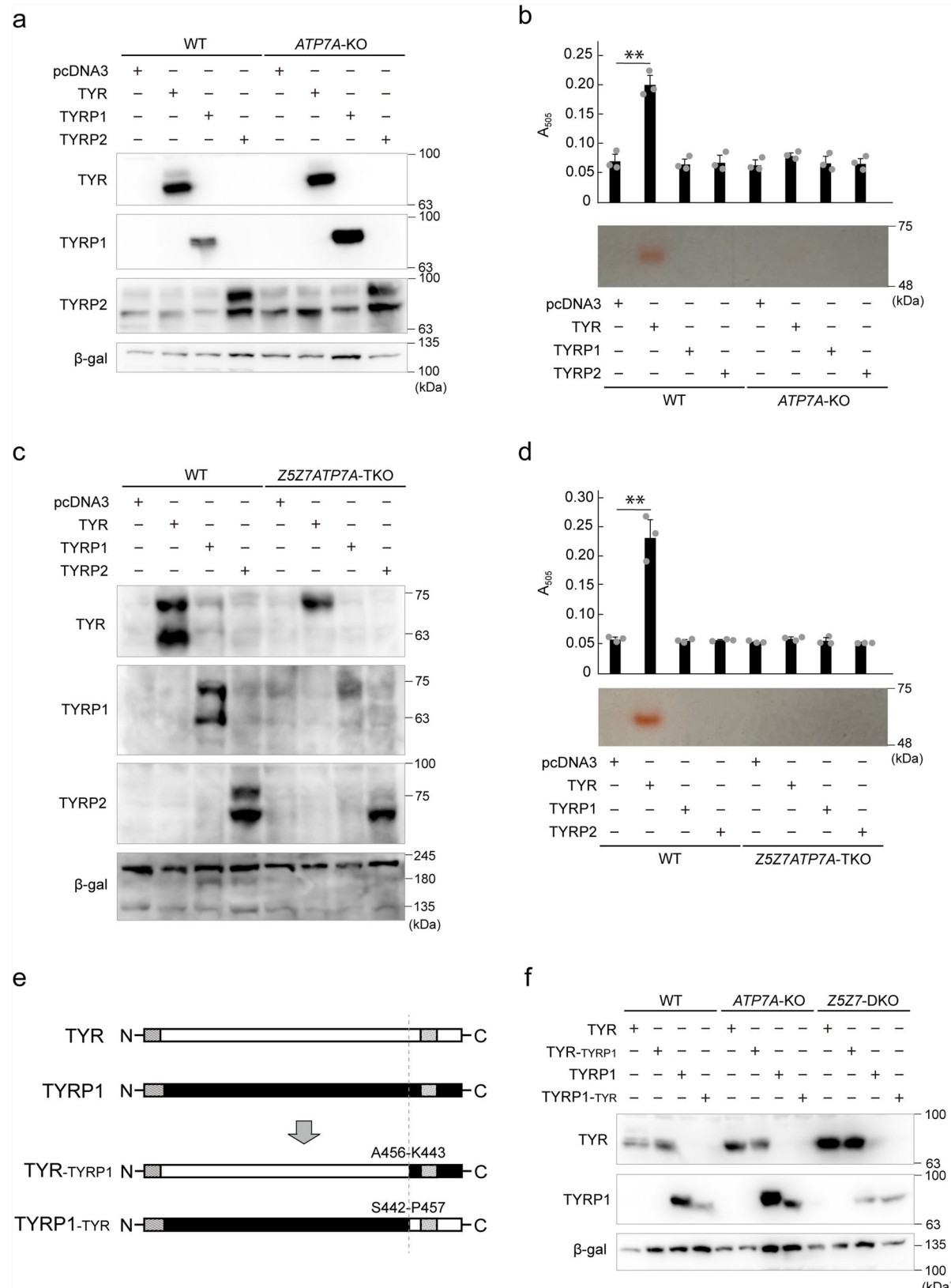

between TYR and TYRP1 and is known to determine intracellular trafficking routes[56]. However, the TM and C-terminal domain of TYRP1 was not sufficient to confer instability to TYR-TYRP1 chimera in the absence of ZNT5-6 and ZNT7, suggesting that additional information contained in the core luminal domain of TYRP1 may be required for metal discrimination (see Fig. 7e, f).

Considering these results and the previous study[58], it is clear that the expression and activation of TYR, TYRP1, and TYRP2 via both by zinc and copper transporters is likely to be highly orchestrated both spatially and temporally, and a complete understanding will require additional studies using protein chimeras and other experimental approaches[59].

**Fig. 7 TYRP1 expression is not dependent on the ATP7A copper transporter. a** Immunoblot analysis shows that the expression of TYRP1, TYR, and TYRP2 in SK-*ATP7A*-KO cells was comparable with levels in WT SK-MEL-2 cells. **b** TYR activity was substantially decreased in SK-*ATP7A*-KO cells. L-DOPA oxidase activity (*upper* graph, $n = 3$) and tyrosinase zymography (*lower* panel) were performed. **c, d** Loss of TYRP1 expression in SK-*Z5Z7*-DKO cells (**c**) and loss of TYR activity in SK-*ATP7A*-KO cells (**d**) were not altered in SK-*Z5Z7ATP7A*-TKO cells. **e** Schematic representation of the chimeric constructs analyzed in **f**. **f** Domain exchange analysis between TYR and TYRP1. TYRP1-$_{TYR}$ chimera mutant showed the same property as that of TYRP1 expressed in SK-*ATP7A*-KO and SK-*Z5Z7*-DKO cells. In **a, c**, and **f**, β-galactosidase (β-gal) was used as a transfection control. In **b** and **d**, statistical significance was determined using the one-way analysis of variance (ANOVA) followed by Tukey's post hoc test. **\*\***$p < 0.01$. Each experiment was performed at least three times, and representative results from independent experiments are shown.

In conclusion, our results reveal that the expression of TYRP1 requires functional zinc transporters ZNT5–6 and ZNT7, thus, demonstrating a requirement for zinc in melanogenesis. These findings provide a marked contrast to TYR, which requires ATP7A dependent delivery of copper. This information sheds light on the maturation process of the TYR family proteins and facilitates our understanding of melanogenesis.

## Materials and methods

**Generation of KO medaka.** Medaka were maintained in an aquarium with recirculating water under a 14/10 h light/dark cycle at 26 °C. The conditions for handling medaka complied with the Regulation for Animal Experiments at Kyoto University, approved by the Animal Research-Animal Care Committee of Kyoto University (R3-45 and Lif-K21023). Detailed information on the editing of *Znt5* and *Znt7* using CRISPR/Cas9 in medaka has been previously described[33]. Briefly, to generate *Znt5*-deficient medaka, mCherry gene driven by the mouse *gamma F* crystalline promoter was inserted into the 5′-region encoding the zinc-binding HDHD motif in exon11 of *Znt5* using homologous recombination. In *Znt7*-deficient medaka, the EGFP gene, driven by the mouse *gamma F* crystalline promoter, was inserted into the sequence encoding the zinc-binding HDHD motif in exon 3 of *Znt7* using homologous recombination. Injected embryos (G0) were cultured and bred into adults. The G0 fish were mated with their wild-type counterparts, and F1 individuals with red (for Znt5) or green (for Znt7) fluorescence in the eyes were selected. The F1 progeny were backcrossed with their wild-type counterpart again, and the resulting F2 individuals, heterozygous for each Znt, were selected. In addition, by mating F1 individuals with red or green fluorescence and continuous mating between their descendants, individuals with various genotype combinations were generated. Each genotype was confirmed via PCR using KOD-FX (Toyobo, Osaka, Japan). The primers used were 5′- CTTCTATTTCCTGTGCCTGAATCT GG-3′ and 5′- CCATAGGAGAAGATCCTGGTG-3′ for *Znt5*, and 5′- GTTCATA CTGGTGTTGGACTCTTGCA-3′ and 5′- CCGTAGGAGAAGCTGTCGTTGGA C-3′ for *Znt7*.

**Cell culture.** Mewo human melanoma cells (JCRB0066, Japanese Collection of Research Bioresources (JCRB) cell bank, Osaka, Japan; deposited by M. Akiyama) were maintained at 37 °C in a humidified 5% CO$_2$ incubator; the culture medium used was EMEM (FUJIFILM Wako Pure Chemical Corporation, Osaka, Japan), containing 10% heat-inactivated FCS (Biosera, Kansas City, MO), 100 U/mL penicillin, and 100 μL/mL streptomycin (Nacalai Tesque, Kyoto, Japan). RPMI1640 (FUJIFILM Wako Pure Chemical Corporation) was used to maintain SK-MEL-2 cells (JCRB1393, JCRB cell bank) as described previously[33]. To inhibit protein degradation, SK-*Z5Z7*-DKO cells were treated for 6 h with 10 or 20 μM MG132 (Peptide Institute, Osaka, Japan) or 10 or 20 nM bafilomycin A1 (Sigma) before collecting the cells. To generate a zinc-deficient culture medium, FCS was treated with Chelex-100 resin as described previously[60]. For zinc supplementation experiments, a cell culture medium containing 75 μM ZnSO$_4$ was used.

**Plasmid construction.** Plasmids used for the expression of hemagglutinin (HA)-tagged human ZNT5 (HA-ZNT5) or ZNT7-HA and their zinc transport-incompetent mutants (HA-ZNT5$_{H451A}$ or ZNT7$_{H70A}$-HA) have been described previously[33]. Mouse *Znt5* cDNA was purchased from Kurabo Bio-Medical Department (Osaka, Japan). The fragment containing *Znt7* cDNA used in this study was PCR-amplified using mouse cDNA prepared from liver total RNA as a template. *TYRP1*, *TYRP2*, and mouse *Tyrp1* cDNAs were purchased from DNA-FORM (Tokyo, Japan). Chicken Tyrp1 was synthesized as a gBlock synthetic gene fragment (Integrated DNA Technologies, Coralville, Iowa, USA). Mouse *Znt5* and *Znt7* genes were FLAG-tagged at their N-terminus, and chicken Tyrp1 was HA-tagged at its C-terminus using two-step polymerase chain reaction (PCR) employing KOD-PLUS (Toyobo)[61]. All cDNAs, including TYR[17], were inserted into the pcDNA3 or pApuro vector[33]. Plasmids expressing chicken Znt5 and Znt7 have been described previously[31]. TYR-$_{TYRP1}$ or TYRP1-$_{TYR}$ chimeras were generated using two-step PCR, as described previously[62], wherein the C-terminal portion of TYR was substituted with that of TYRP1, and vice versa in the latter (TYR-$_{TYRP1}$, in which TYR (Met1 to Ser442) was fused with TYRP1 (Pro457 to Val537), and TYRP1-$_{TYR}$, in which TYRP1 (Met1 to Ala456) was fused with TYR

(Lys443 to Luu529)). Plasmids expressing β-galactosidase (pβactβgal) have been previously described[63].

**Disruption of TYRP1, ZNT5, and ZNT7 in Mewo cells and of ATP7A in SK-MEL-2 cells.** *ATP7A* and *TYRP1* were disrupted using CRISPR/Cas9-mediated genome editing. Oligonucleotides for generating sgRNA expression plasmids were annealed and cloned into the BsaI site of the PX-330-B/B vector[31]. The following oligonucleotides were used for guide RNA (gRNA) for *ATP7A*: ATP7A-F: 5′- CA CCGGTGAAGAGTTGCAAAGTGG−3′, ATP7A-R: 5′- AAACCCACTTTGCAA CTCTTCACC-3′, *TYRP1*: TYRP1-F: 5′- CACCGAGCTTCATCCATACTGCGTC −3′, TYRP1-R: 5′- AAACGACGCAGTATGGATGAAGCTC-3′ (gRNA sequences are underlined). The oligonucleotides used for the gRNAs of *ZNT5* and *ZNT7* were cloned into the PX-330-B/B vector, as described previously[31]. The constructed plasmids and one-tenth quantity of pcDNA6/TR (which contains the blasticidin S resistance gene) or pA-puro (which contains the puromycin resistance gene) were cotransfected into 80% confluent Mewo or SK-MEL-2 cells using Lipofectamine 2000 reagent in Opti-MEM medium (Invitrogen). The cells were cultured for 1 day and transferred to a 10 cm cell culture dish containing the culture medium with 2 or 20 μg/mL blasticidin S (InvivoGen, San Diego, CA, USA) for Mewo and SK-MEL-2 cells, or 0.75 μg/mL puromycin (Sigma) for Mewo cells to establish stable clones. Mewo and SK-MEL-2 cells deficient in *ATP7A*, *TYRP1*, or *ZNT5* and *ZNT7* were confirmed by direct sequencing of the fragments amplified using genomic DNA PCR (Table 1). The primers used are listed in Supplementary Table 1.

**Transient and stable transfection.** Each expression plasmid was transiently transfected into 80% confluent SK-MEL-2 cells using Lipofectamine 2000 reagent in Opti-MEM (Invitrogen). After 4 h, fresh medium was added and the cells were cultured for 24 h and used for immunoblotting or immunofluorescence staining. In some experiments, the amounts of expression plasmids for human TYR, TYRP1, TYRP2, mouse Tyrp1, or chicken Tyrp1 and those of ZNT5, ZNT7, or ATP7A were set at 1:10. For the experiments to measure the TYR activity, the amounts of expression plasmids of TYR, TYRP1, or their chimera mutants and that of pβactβgal were set at 10:1. For stable transfection to express ZNT7 in Mewo cells, after transfection, the cells were cultured in the presence of 0.75 μg/mL puromycin (Sigma) to establish stable transformants.

**Preparation of membrane protein.** Membrane proteins were obtained from the pelleted cells and resuspended in 1.4 mL of cold homogenizing buffer (0.25 M sucrose, 20 mM HEPES, and 1 mM EDTA) and homogenized with 30 strokes of a 7 mL Dounce homogenizer. To remove the nuclei, the homogenate was centrifuged at 5,000 rpm for 5 min at 4 °C. The post-nuclear supernatant was centrifuged at 15,000 rpm for 30 min at 4 °C. The pellet was washed once with TBS and lysed either in NP40 lysis buffer (100 mM NaCl, 50 mM HEPES, 1% NP40) to analyze TYR, TYRP1, and TYRP2 expression, or with tyrosinase lysis buffer (60 mM Tris-HCl (pH 6.0), 150 mM NaCl, 1.0% Triton-X) for tyrosinase zymography. To prepare total cellular lysates, pelleted cells were washed once with PBS and lysed in NP40 lysis buffer. Protein concentration was determined using a DC protein assay (Bio-Rad, Hercules, CA, USA) with bovine serum albumin as a standard.

**Immunoblotting analysis.** Immunoblotting was performed as described previously[64]. Briefly, membrane proteins (20 μg) or total cellular lysate (20 μg) were lysed in 6× SDS sample buffer at 95–100 °C for 3 min or at 37 °C for 30 min (for detection of ZNTs), separated using electrophoresis on SDS polyacrylamide gels, and transferred onto PVDF membranes (Immobilon-P, Millipore Corp., Bedford, MA, USA). Blotted PVDF membranes were blocked with 5% skimmed milk/0.1% Tween-20 in PBS and then incubated with the following primary antibodies (diluted in blocking buffer): anti-HA [HA-11] (1:3000; BioLegend, San Diego, CA), anti-HA [3F10] (1:3000, Roche Molecular Biochemicals, Germany), anti-DDDDK-Tag [PM020] (1:3000; MBL), anti-DYKDDDDK tag [20543-1-AP] (1:3000; Proteintech Group Inc., Chicago, IL), anti-ZNT5 (1:500)[65], anti-TYRP1 [G-7] (sc-166857), anti-TYRP1 [ab235446] (1:1000; Abcam), anti-TYRP1 [HPA00937] (1:3000; Sigma-Aldrich), anti-TYRP1 [ab235447] (1:1000; Abcam for detecting mouse Tyrp1), anti-TYR [T311] (1:3000; Santa Cruz Biochemistry), anti-TYRP2 [C-9] (1:3000; Santa Cruz Biochemistry), anti-α-tubulin [12G10] (1:3000; deposited to Developmental Studies Hybridoma Bank (DSHB) by J. Frankel and E. M. Nelsen), anti-calnexin [10427-2-AP] (1:3000; Proteintech), and anti-β-gal

[M203-3] (1:3000; MBL Nagoya, Japan). Immunoreactive bands were detected using 1:3000 horseradish peroxidase-conjugated anti-mouse, anti-rabbit, or anti-rat secondary antibodies (NA931, NA934, or NA935, Cytiva, Marlborough, MA), or anti-goat secondary antibody (sc-2020, Santa Cruz Biotechnology), and Immobilon Western Chemiluminescent HRP Substrate (Millipore) or SuperSignal West Atto Ultimate Sensitivity Substrate (Pierce, Rockford, IL, USA). Chemiluminescent images were obtained using ImageQuant LAS 500 (Cytiva). Full-length immuno-blot images are provided in Supplementary Figs. 1–5.

**Immunofluorescence staining**. Mewo cells were cultured on coverslips and fixed in 10% formaldehyde neutral buffer solution (Nacalai Tesque). For SK-MEL-2 cells, coverslips were coated with 0.01% poly-L-lysine (Sigma-Aldrich). Immunostaining was performed as described previously[31] using the following first antibodies: anti-TYR [T311] (1:1000; Santa Cruz Biochemistry), anti-TYRP1 [ab235446] (1:1000; Abcam), anti-TYRP1 [ab235447] (1:1000; Abcam for detecting mouse Tyrp1), anti-TYRP2 [C-9] (1:1000; Santa Cruz Biochemistry), anti-HA [3F10] (1:3000, Roche Molecular Biochemicals), and anti-FLAG [M2] (1:3000; F3165, Sigma). Goat anti-mouse IgG conjugated to Alexa 488, goat anti-rabbit IgG conjugated to Alexa 488, donkey anti-goat IgG conjugated to Alexa 488, goat anti-rabbit IgG conjugated to Alexa 594, and goat anti-rat IgG conjugated to Alexa 594 (Abcam) were used as the second and third antibodies. The antibodies were applied at room temperature for 1 h or at 4 °C overnight, and 4′,6-diamino-2-phenylindole (DAPI; 1:1000; Abcam) was added during second and third antibody staining to label nuclei. After three PBS washes, the coverslips were mounted onto glass slides using the SlowFade Diamond Antifade Mountant reagent (Thermo Fisher Scientific). The stained cells were examined using a fluorescence microscope and a ×60 magnification oil-immersion objective (FSX100; Olympus, Tokyo, Japan). Identical exposure settings and times were used for the corresponding images shown in each figure.

**Tyrosinase zymography**. SK-MEL-2 cells, transiently transfected with expression plasmids of TYR, TYRP1, or their chimeric mutants, were lysed in tyrosinase lysis buffer (60 mM Tris-HCl (pH 6.8), 150 mM NaCl, 1.0% Triton-X), and then lysed in 6× SDS buffer (375 mM Tris-HCl (pH6.8), 30% glycerol, 6% SDS, 0.1% BPB) without reducing agents, before electrophoresis. Samples were separated on an 8% polyacrylamide SDS gel at 4 °C, and gels were equilibrated in 50 mM phosphate buffer (pH 6.0) for 3 h with gentle shaking. Colorimetric staining was carried out by incubating gels with a color formation buffer (10 mM) in phosphate buffer (pH 6.8) containing 1.5 mM L-DOPA (Nacalai Tesque) and 4 mM 3-methyl-2-benzothiazolinone hydrazone hydrochloride hydrate (MBTH, Tokyo Kasei Kogyo Co., Ltd., Tokyo, Japan) for 1 h at 37 °C.

**Measurements of TYR activity (L-DOPA oxidase activity)**. Total cell lysates were prepared by lysing transfected SK-MEL-2 cells in tyrosinase lysis buffer (60 mM Tris-HCl (pH 6.8), 150 mM NaCl, and 1.0% Triton-X) were used. Two hundred microliter pre-reaction mix (5 mM L-DOPA, 20.7 mM 3-methyl-2-benzothiazolinone hydrazone hydrochloride hydrate (MBTH), 50 mM phosphate buffer, pH 7.1, 2% dimethylformamide) was added to the lysate, and the mixture was incubated for 1 h at 37 °C after gentle shaking for 5 min. The reaction was monitored by measuring the increase in color intensity at 505 nm using a Synergy H1 Hybrid multi-mode microplate reader (BioTek, Winooski, VT, USA).

**Electron microscopy**. TEM observations were performed at the Hanaichi Electron Microscope Technical Laboratory (Aichi, Japan). Medaka (WT, $Znt5^{+/-}:Znt7^{-/-}$, or $Znt5^{-/-}:Znt7^{-/-}$) was fixed in 0.1 M phosphate-buffered 2% glutaraldehyde, and then post-fixed in 2% osmium tetroxide for 3 h in an ice bath. The specimens were then stained with 0.5% aqueous uranyl acetate for 1 h, dehydrated in graded ethanol, and embedded in an epoxy resin. To observe melanosomes in WT Mewo, Mewo-Z5Z7-DKO, and Mewo-TYRP1-KO cells, which were cultured for 11 days (the culture medium was replaced every 3 days), the cells cultured in 6 cm dishes were fixed and treated in the same manner. Ultrathin sections were cut using an ultramicrotome, stained with uranyl acetate for 15 min and modified Sato's lead solution for 5 min, and then examined using TEM (JEM-1200EX, JEOL, Tokyo, Japan).

**Quantitation of melanin**. Melanin content was measured as described elsewhere[66,67], with a small modification. Mewo cells cultured for 11 days (the culture medium was replaced every 3 days) were collected and washed with Tris-buffered saline (TBS). After centrifugation, the precipitated cells were solubilized by treatment with 500 μL of 1 M NaOH aqueous solution at 55 °C for 1 h in test tubes. Absorbance was measured at a wavelength of 405 nm. To measure the melanin content in medaka fish, whole bodies of four medaka, after removing the head with a razor blade, were grouped ($n = 3$). A standard curve was generated using melanin (M0418, Sigma-Aldrich) dissolved in 1 M NaOH. Melanin content per cell was calculated using the protein content.

**RT-PCR**. Total RNA was isolated from harvested Mewo cells using Sepasol I (Nacalai Tesque, Kyoto, Japan) and reverse transcribed using ReverTra Ace (Toyobo, Osaka, Japan). PCR was performed using KOD-FX (TOYOBO); the primers used are shown in Supplementary Table 2. Full-length RT-PCR image is provided in Supplementary Fig. 1f.

**Statistics and reproducibility**. All data are expressed as the mean ± SD of tri-plicate experiments. Statistical significance was determined using one-way analysis of variance (ANOVA) followed by Tukey's post hoc test (for comparing three or more groups) or Student's $t$-test (for comparing two groups). Differences were considered significant at $p < 0.01$ (**). All experiments were repeated at least three times.

## Data availability

All data generated or analyzed during this study are included in this published article and its Supporting Information files (Full-length immunoblot and RT-PCR images are provided in Supplementary Figs. 1–5, and the data used to create all figure graphs is provided in Supplementary Data 1) or are available from the corresponding author upon reasonable request.

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

## Acknowledgements

We thank Yayoi Kurokawa for technical assistance. This work was supported by a Grant-in-Aid for Scientific Research on Innovative Areas "Integrated Bio-metal Science" (MEXT KAKENHI Grant Number JP19H05768) from the Ministry of Education, Culture, Sports, Science, and Technology, Japan; Grant-in-Aid for Scientific Research (B) (JSPS KAKENHI Grant Number JP22H02257) from the Japan Society for the Promotion of Science; the KOSÉ Cosmetology Research Foundation; the Kao Melanin Workshop; Lydia O'Leary Memorial Pias Dermatological Foundation; and the Nagase Science and Technology Foundation (to T.K.).

## Author contributions

T.K. designed the study; T.W., E.S., M.S., A.S., Y.N., M.K., and T.K. collected, analyzed, and interpreted the data; H.A. and H.H. interpreted the data; M.J.P. provided technical assistance; and T.K. drafted the manuscript. All authors have reviewed the manuscript.

## Competing interests
The authors declare no competing interests.
