## [Peer Review File · Communications Biology]

Reviewers' comments:

Reviewer #1 (Remarks to the Author):

General comment

I think this is a very valuable contribution to the determination of the metal cofactor in the tyrosinase family (tyrosinase and two tyrosinase related proteins, TRP1 and TYRP2). The presence of copper at the tyrosinase active site and Zinc at TRP2 are clear, but the nature of the metal cofactor at the TRP1 has been elusive for years. Authors include in the manuscript most of the important references to support this starting situation. I would suggest (I excuse about suggestions of new references, but I think that the inclusion of the paper entitled "On the metal cofactor in the tyrosinase family. International journal of molecular sciences, 19(2), 633 is very pertinent in the context of this field.

Briefly, the DHICA oxidase activity proposed to TRP1 would suggest copper (copper shows redox properties), but more recent structural finding proposed that TRP1 contains Zinc, and this metal does not reconcile with any oxidase activity. Under my opinion, this manuscript clearly shows that the metal ion is Zinc and this point is a clear contribution to progress in this aspect. To me, this is the most important contribution to the knowledge of the tyrosinase family, as this particular point clear up the current ambiguity. As consequence, the (catalytic, tyrosinase-stabilizing protein or a second decarboxylating dopachrome tautomerase) function of TRP1 and the inclusion of all the tyrosinase family as type 3 copper proteins would be reconsidered. It seems that only tyrosinase is a type 3 copper protein, but the paralogous proteins are not. All results included in the manuscript are well performed and they consistently indicate that the requirement of Zinc transporters for TYRP1 expression is an intrinsic property of TYRP1, and that the copper transporter, ATP7A, does not affect the expression of TYRP1. Techniques are appropriate and convincing, and the figures are excellent illustration of the results. Discussion and conclusions are correct. References are also appropriate and abundant. On this regard, discussion about chimeras is also well performed, but a reference to the paper Olivares et al. Conformation-dependent post-translational glycosylation of tyrosinase: Requirement of a specific interaction involving the CuB metal binding site. Journal of Biological Chemistry, 278(18), 15735-15743 would enrich it.

Minor points:

Correct some grammatical errors, at least the following: pigmentation (abstract), phenotype (line 96), melanomes (by melanosomes, line 120), lysosomal (line 246), indespicable (line 261), lumenal (line 282), requirment (line 289), Anaysis (line 787), immunofluorescence (line 790)
Line 60-61: Dopaquinone is then isomerized to dihydroxyindole carboxylic acid (DHICA) via transformation of dopachrome by TYRP2 (dopachrome tautomerase) 10.
Please replace "isomerized" by "transformed", as DQ and DHICA are not isomers.

Reviewer #2 (Remarks to the Author):

Pigmentation and TYRP1 expression are mediated by zinc through the early secretory pathway-resident ZNT proteins

Takumi Wagatsuma, Eisuke Suzuki, Miku Shiotsu, Akiko Sogo, Yukina Nishito, Hideya Ando, Hisashi Hashimoto, Michael J. Petris, Masato Kinoshita and Taiho Kambe

Wagatsuma and colleagues explore the role of zinc transporters ZnT5 and ZnT7 in the function of the enzymes TYRP1 in pigmentation with a range of in vivo and in vitro experimental strategies.

This study investigates an interesting question addressing the role of zinc in the activity of zinc-binding enzymes mediated by the zinc transporters ZnT5 and ZnT7. Zinc transporters of the ZnT family are localised in the membrane of secretory compartments and were shown to be involved in Zn uptake in intracellular vesicles. Previous studies conducted by the same authors' group, showed a link between zinc transporters of the ZnT family and the activity of Alkaline Phosphatase.

Comments and suggestions

Introduction -

The introduction provides the state of arts and context of melanogenesis, it would be beneficial to provide some context about the role of zinc and its membrane transporters in biology and more specifically in the biosynthesis and activation of metal enzymes.

Is the role of zinc or any other zinc transporters been investigated in melanogenesis?

Are ZnT5 and ZnT7 expressed in medaka larvae and human melanocytes and is any other zinc transporters expressed in those cells?

Results -

Zinc transporter ZnT5 and ZnT7 mutants were developed in medaka larvae using CRISPR-Cas9 $Znt5^{-/-}/Znt7^{-/-}$, and $Znt5^{-/+}/Znt7^{-/-}$ to demonstrate the role of the corresponding proteins in melanin synthesis. How are the authors rule out that ZnT7 is irrelevant in melanogenesis?

Q 1 from figure 1 it is clear that Znt5 was required for melanin synthesis, however the role of ZnT7 was not demonstrated. The authors should demonstrate the presence and activity of ZnT7 homodimer and ZnT5/ZnT6 heterodimer in the system used in the set of experiments presented. The functional experiments including also $Znt5^{-/-}/Znt7^{-/+}$ mutants should be included in the analysis.

Q 2 the authors should demonstrate that both ZnT7 homodimer and ZnT5/ZnT6 heterodimer are required in the human melanoma Mewo cell line for melanin synthesis. Can the authors demonstrate that ZnT5 is only active as part of the ZnT5/ZnT6 dimer?

Q 3 was the melanin phenotype investigated in loss-of-function mutants of other zinc transporters genes?

Q 4 The authors demonstrated that zinc deficiency does not affect the expression of TYRP1, however, TYRP1 expression is dependent on ZnT5/ZnT7. It would be beneficial to demonstrate whether zinc deficiency affects melanin production.

Reviewer #3 (Remarks to the Author):

Wagatsuma and colleagues provide compelling evidence that melanogenesis is dependent on the zinc transporters ZNT5-6 and ZNT7. In addition, they show that, of the three melanogenic enzymes tyrosinase (TYR) and tyrosinase-related proteins 1 and 2 (TYRP1 and TYRP2), expression and function of TYRP1 is dependent on functional ZNT5-6 and ZNT7. This is in contrast to TYR which requires ATP7A-dependent supply of copper. Loss of ZNT5-6 and ZNT7 function resulted in hypopigmentation in medaka fish and human melanoma cells, and was associated with immature melanosomes and decreased melanin content, and greatly reduced TYRP1 expression. The requirement of ZNT5-6 and ZNT7 for expression and function of TYRP1 is conserved in human, mouse and chicken orthologs.

The present study is groundbreaking in that it demonstrates the importance of both zinc and copper transporters in melanin synthesis. Combined with previously published crystallographic evidence that

zinc ions are present in the active site of TYRP1, the current findings support a model in which TYRP1 activity and stability requires binuclear zinc in its active site to be delivered by ZNT5-6 and ZNT7.

This is a highly novel and seminal result, strongly supporting the role of zinc in melanogenesis. The research has been carried out competently, and the manuscript is well-written and clear. It was a joy reading it.

I have only a few minor textual comments:

Line 41: pigmentation -> pigmentation

Line 77: make zinc -> supply zinc (?)

Line 261: indispensible -> indispensable

Line 303: crystalline promoter -> alpha-crystallin promoter? beta-crystallin promoter? Also change in line 305.

Line 787: anaysis-> analysis

Line 801: TYPR1 -> TYRP1

Response to Reviewers' Comments

Reviewer 1:

I think this is a very valuable contribution to the determination of the metal cofactor in the tyrosinase family (tyrosinase and two tyrosinase related proteins, TRP1 and TYRP2). The presence of copper at the tyrosinase active site and Zinc at TRP2 are clear, but the nature of the metal cofactor at the TRP1 has been elusive for years. Authors include in the manuscript most of the important references to support this starting situation. I would suggest (I excuse about suggestions of new references, but I think that the inclusion of the paper entitled "On the metal cofactor in the tyrosinase family. International journal of molecular sciences, 19(2), 633 is very pertinent in the context of this field.

Briefly, the DHICA oxidase activity proposed to TRP1 would suggest copper (copper shows redox properties), but more recent structural finding proposed that TRP1 contains Zinc, and this metal does not reconcile with any oxidase activity. Under my opinion, this manuscript clearly shows that the metal ion is Zinc and this point is a clear contribution to progress in this aspect. To me, this is the most important contribution to the knowledge of the tyrosinase family, as this particular point clear up the current ambiguity. As consequence, the (catalytic, tyrosinase-stabilizing protein or a second decarboxylating dopachrome tautomerase) function of TRP1 and the inclusion of all the tyrosinase family as type 3 copper proteins would be reconsidered. It seems that only tyrosinase is a type 3 copper protein, but the paralogous proteins are not. All results included in the manuscript are well performed and they consistently indicate that the requirement of Zinc transporters for TYRP1 expression is an intrinsic property of TYRP1, and that the copper transporter, ATP7A, does not affect the expression of TYRP1. Techniques are appropriate and convincing, and the figures are excellent illustration of the results. Discussion and conclusions are correct. References are also appropriate and abundant. On this regard, discussion about chimeras is also well performed, but a reference to the paper Olivares et al. Conformation-dependent post-translational glycosylation of tyrosinase: Requirement of a specific interaction involving the CuB metal binding site. Journal of Biological Chemistry, 278(18), 15735-15743 would enrich it.

Response: Thank you for appreciating our work and providing helpful suggestions. We have already included the suggested article of "International journal of molecular sciences, 19(2), 633" as Ref. No. 2 in the original manuscript. We have added "Journal of Biological Chemistry, 278(18), 15735-15743" as Ref. No. 58 in Discussion (line 287).

1) Correct some grammatical errors, at least the following: pigmentation (abstract), phenotype (line 96), melanomes (by melanosomes, line 120), lysosomal (line 246), indispensable (line 261), luminal (line 282), requirement (line 289), Analysis (line 787), immunofluorescence (line 790)

Line 60-61: Dopaquinone is then isomerized to dihydroxyindole carboxylic acid (DHICA) via transformation of dopachrome by TYRP2 (dopachrome tautomerase) 10. Please replace “isomerized” by “transformed”, as DQ and DHICA are not isomers.

Response: We apologize for these grammatical errors. We have corrected them at the corresponding lines in the revised manuscript.

Reviewer 2:

Introduction -

The introduction provides the state of arts and context of melanogenesis, it would be beneficial to provide some context about the role of zinc and its membrane transporters in biology and more specifically in the biosynthesis and activation of metal enzymes.

Is the role of zinc or any other zinc transporters been investigated in melanogenesis?

Are ZnT5 and ZnT7 expressed in medaka larvae and human melanocytes and is any other zinc transporters expressed in those cells?

Results -

Zinc transporter ZnT5 and ZnT7 mutants were developed in medaka larvae using CRISPR-Cas9 $Znt5^{-/-}/Znt7^{-/-}$, and $Znt5^{-/+}/Znt7^{-/-}$ to demonstrate the role of the corresponding proteins in melanin synthesis. How are the authors rule out that ZnT7 is irrelevant in melanogenesis?

Response: Thank you for the important comments. We have answered to these questions specifically as follows.

1) from figure 1 it is clear that Znt5 was required for melanin synthesis, however the role of ZnT7 was not demonstrated. The authors should demonstrate the presence and activity of ZnT7 homodimer and ZnT5/ZnT6 heterodimer in the system used in the set of experiments presented. The functional experiments including also Znt5^{-/-}/Znt7^{-/+} mutants should be included in the analysis.

Response: We recently reported that the photos of WT medaka and Znt5^{+/-};Znt7^{+/-}, Znt5^{+/-};Znt7^{-/-}, Znt5^{-/-};Znt7^{+/-}, and Znt5^{-/-};Znt7^{-/-} medaka mutants, as well as the movies of Znt5^{+/-};Znt7^{+/-}, Znt5^{+/-};Znt7^{-/-}, and Znt5^{-/-};Znt7^{+/-} medaka mutants (Ref. No. 33 in the revised manuscript). We did not mention the defects of pigment in the article, but the photos and movies clearly showed both Znt5^{+/-};Znt7^{-/-} and Znt5^{-/-};Znt7^{+/-} medaka mutants have intact pigmentation. The reason for using Znt5^{+/-};Znt7^{-/-} instead of Znt5^{-/-};Znt7^{-/+} mutant medaka in this study is because Znt5^{-/+};Znt7^{-/-} mutants did not show any apparent defects, whereas Znt5^{-/-};Znt7^{-/+} mutant medaka showed touch-insensitive phenotypes (This is the research focus in Ref. No. 33 in the revised manuscript). We may omit the photo and data of Znt5^{-/+};Znt7^{-/-} mutant medaka from Fig.1, however, the data in other figures clearly show the involvement of ZNT7 in melanogenesis. Moreover, we believe they are good as a control data and thus, we have retained them.

In the manuscript (Ref. No. 33 in the revised manuscript) and in our other previous studies (Ref. Nos. 28-32 in the revised manuscript), we have shown that our experiment strategy using ZNT5 and ZNT7 mutants are competent in evaluating ZNT5-6 and ZNT7 activity and functions.

To clarify these points, we have added the brief sentences in Introduction (lines 78-80, 85, and 87-88), and added new references (Ref. Nos. 24, 34, and 35).

2) the authors should demonstrate that both ZnT7 homodimer and ZnT5/ZnT6 heterodimer are required in the human melanoma Mewo cell line for melanin synthesis. Can the authors demonstrate that ZnT5 is only active as part of the ZnT5/ZnT6 dimer?

Response: We reported that ZNT5 is the only active subunit and ZNT6 is functional as an auxiliary subunit using their mutants, which is confirmed by other groups. To clarify these points, we have added the brief sentence in Introduction (lines 87-88) and added new references (Ref. Nos. 33 and 34 in the revised manuscript).

3) was the melanin phenotype investigated in loss-of-function mutants of other zinc transporters genes?

Response: In vertebrates, 24 zinc transporters are operative. Each is thought to have specific cellular functions. Among them, only ZNT5-6 and ZNT7 are localized to the early secretory pathway and supply zinc to zinc proteins, which traffic to the extracellular space and, the plasma membrane, the organelles after zinc metallation, as reported previously (Ref. Nos. 28-34 in the revised manuscript). Therefore, we agree with the possibility that other zinc transporter may be involved in melanin biosynthesis, but we believe it would occur in the mechanisms other than zinc metallation of TYRP1 (e.g. Loss of zinc transporters might impair melanocyte functions, or might disturb melanosome maturation and trafficking of TYRs).

Moreover, we are now investigating how other proteins, which are involved in the cellular zinc metabolism including zinc transporters, are associated with zinc metalation through ZNT5-6 and ZNT7. We think answering this question should be clarified at the next stage, which is beyond the scope of this manuscript.

4) The authors demonstrated that zinc deficiency does not affect the expression of TYRP1, however, TYRP1 expression is dependent on ZnT5/ZnT7. It would be beneficial to demonstrate whether zinc deficiency affects melanin production.

Response: Thank you for the important comments. We showed TYRP1 expression was not impaired in zinc deficient culture for 48 h, whereas we measured melanin contents using Mewo cells cultured for 11 days. Actually, it is impossible to culture the Mewo cells in the same zinc deficient conditions for longer than 48 h, because the conditions are too severe for the Mewo cells to grow. At present, there are no reports clearly revealing that zinc deficiency results in hypopigmentation, even in the case that it causes alopecia (e.g. Genes & Nutrition,1, 61-70, 2006). Pigmentation is affected only when zinc supply through ZNT5-6 and ZNT7 is impaired, as described in this manuscript.

Reviewer 3:

The present study is groundbreaking in that it demonstrates the importance of both zinc and copper transporters in melanin synthesis. Combined with previously published crystallographic evidence that zinc ions are present in the active site of TYRP1, the current findings support a model in which TYRP1 activity and stability requires binuclear zinc in its active site to be delivered by ZNT5-6 and ZNT7.

This is a highly novel and seminal result, strongly supporting the role of zinc in melanogenesis. The research has been carried out competently, and the manuscript is well-written and clear. It was a joy reading it.

Response: Thank you for appreciating our work and providing valuable suggestions.

1) I have only a few minor textual comments:

Line 41: pigmentation -> pigmentation

Line 77: make zinc -> supply zinc (?)

Line 261: indispensible -> indispensable

Line 303: crystalline promoter -> alpha-crystallin promoter? beta-crystallin promoter? Also change in line 305.

Line 787: anaysis-> analysis

Line 801: TYPR1 -> TYRP1

Response: We apologize for these grammatical errors and insufficient explanation. We have corrected and modified them at the corresponding lines in the revised manuscript.

REVIEWERS' COMMENTS:

Reviewer #2 (Remarks to the Author):

The authors responded to the reviewers' comments with convincing arguments. In my opinion, the manuscript can now be published.